# Solving a System of Differential Equations Containing a Diffusion Equation with Nonlinear Terms on the Example of Laser Heating in Silicon

**Vladimir Lipp [1,2,3,]\***, **Baerbel Rethfeld [1]**, **Martin Garcia [2]** and **Dmitry Ivanov [1,2,4]**

[1] Department of Physics and OPTIMAS Research Center, Technical University of Kaiserslautern, 67663 Kaiserslautern, Germany; rethfeld@physik.uni-kl.de

[2] Institute of Physics and Center for Interdisciplinary Nanostructure Science and Technology (CINSaT), University of Kassel, 34125 Kassel, Germany; garcia@physik.uni-kassel.de (M.G.); ivanov@uni-kassel.de (D.I.)

[3] Center for Free-Electron Laser Science CFEL, Deutsches Elektronen-Synchrotron DESY, Notkestr. 85, 22607 Hamburg, Germany

[4] P. N. Lebedev Physical Institute of Russian Acad. Sci., 119991 Moscow, Russia

**\*** Correspondence: vladimir.lipp@desy.de

**Abstract:** We present a finite-difference integration algorithm for solution of a system of differential equations containing a diffusion equation with nonlinear terms. The approach is based on Crank–Nicolson method with predictor–corrector algorithm and provides high stability and precision. Using a specific example of short-pulse laser interaction with semiconductors, we give a detailed description of the method and apply it to the solution of the corresponding system of differential equations, one of which is a nonlinear diffusion equation. The calculated dynamics of the energy density and the number density of photoexcited free carriers upon the absorption of laser energy are presented for the irradiated thin silicon film. The energy conservation within $0.2\%$ has been achieved for the time step $10^8$ times larger than that in case of the explicit scheme, for the chosen numerical setup. The implemented Fortran source code is available in the Supplementary Materials. We also present a few examples of successful application of the method demonstrating its benefits for the theoretical studies of laser–matter interaction problems. Finally, possible extension to 2 and 3 dimensions is discussed.

**Keywords:** finite-difference scheme; diffusion equation; laser–matter interaction; semi-implicit algorithm; system of partial differential equations; ultrashort lasers

## 1. Introduction

Many phenomena occurring in nature for their investigation can be described via mathematical models based on time-dependent nonlinear diffusion equations [1]. Examples include genetics [2,3], image processing [4], quantum mechanics [5], and laser–material interactions [6]. Although during the last decades a big effort has been undertaken to find efficient numerical schemes for solution of the corresponding mathematical problem, some of the applications are still a challenging task. Specifically, the efforts in model implementation as well as their demands on the computational power during processing can substantially hinder the theoretical interpretation of the investigated problem. In this work, we consider an application of the nonlinear parabolic diffusion equation to describe the response of solids to an ultrashort laser pulse irradiation. Apart from the insights into the material structure, this topic is important for the description of laser machining [7–9] and nanostructuring experiments [10–12] with applications in biotechnology [13] and IT [14,15]. For metals,

the problem may be mathematically formulated in the form of frequently used Two-Temperature Model (TTM) [16], whereas for semiconductors, a similar TTM-like approach has been proposed [17]. The latter is based on the system of partial differential equations, reflecting the conservation laws in the atomic subsystem of a solid and its electronic subsystem. Although it is relatively simple to apply an explicit finite-difference numerical scheme to solve such systems in metals [18] or semiconductors [19], the corresponding stability criteria demand the integration time steps to be small, causing high computational costs as a result. The main restriction on the time step often comes from the nonlinear diffusion equation describing the carrier heat conduction process [20]. One of the possibilities to increase the time step of diffusion equation is to use implicit or semi-implicit integration schemes. For instance, the Crank–Nicolson semi-implicit scheme [21,22] provides unconditionally stable solution when applied to linear diffusion equations. However, this approach is not directly applicable when nonlinear terms play an important role.

In this work, we present a semi-implicit finite-difference method for the solution of a system of differential equations—one of which is a diffusion equation with nonlinear terms—and apply it to model short laser pulse interaction with silicon. The presented approach is based on Crank–Nicolson method with predictor–corrector algorithm and provides high stability and precision. It has been already successfully applied for the investigation of ultrashort laser interaction with metals [23] and semiconductors [24–26]. Section 2 describes the theoretical model and presents the system of equations where a nonlinear diffusion equation results in strong restriction on the time step for the explicit integration algorithm. In Section 3, we give a detailed description of semi-implicit numerical solution scheme, which was modified with predictor–corrector algorithm to account for the nonlinearity in the diffusion equation. Further, in Section 4, the calculation results for a particular set of parameters are presented and the energy conservation versus the applied iteration time step is investigated. Section 5 mentions the existing works, in which this approach has been successfully utilized, and suggests possible improvements for the application of the presented approach in two-dimensional (2D) and three-dimensional (3D) cases. Finally, in Section 6, we give a summary of our results.

## 2. Model Description

To demonstrate the application of the finite-difference scheme, we use an example of laser irradiation on silicon. Below, we present the full set of nonlinear differential equations for the continuum description of electron density and electron/phonon energy density dynamics in silicon under ultrashort laser irradiation. For the derivation of the following expressions we refer to the works in [17] and [25]. Due to laser pulse irradiation (in this example Ti:Sapphire laser at 800 nm wavelength), free carriers are generated in the material, electrons in the conduction band, and holes in the valence band. Both types of carriers are assumed to quickly equilibrate in the corresponding bands and move together due to the Dember field preventing charge separation [27]. To each of them, we apply separate Fermi–Dirac distributions with different chemical potentials, $\phi_e$ and $\phi_h$, for the electrons and holes, respectively, but with shared carrier density $n$ and temperature $T_e$ (two-chemical potentials model). The reduced chemical potentials are defined as follows,

$$\eta_e = \frac{\phi_e - E_C}{k_B T_e} \text{ and } \eta_h = \frac{E_V - \phi_h}{k_B T_e}, \tag{1}$$

where $E_C$ and $E_V$ are the conduction and valence band energy levels, respectively, so the energy gap is $E_g = E_C - E_V$. The integration of the carrier distribution functions over the energy leads to the expressions for the carrier density (parabolic bands are assumed):

$$n = 2 \left( \frac{m_c^* k_B T_e}{2\pi\hbar^2} \right)^{\frac{3}{2}} F_{\frac{1}{2}} (\eta_c), \tag{2}$$

where subscript *c* stands as *e* for electrons and *h* for holes. The Fermi–Dirac integral is defined as

$$F_\xi (\eta_c) = \frac{1}{\Gamma (\xi + 1)} \int_0^\infty \frac{x^\xi}{1 + \exp (x - \eta_c)} \, dx. \tag{3}$$

The carrier current is the sum of contributions from the electrons and the holes:

$$
\vec{J} = -D \Bigg[ \nabla n + \frac{n}{k_B T_e} \left[ H^{\frac{1}{2}}_{-\frac{1}{2}} (\eta_e) + H^{\frac{1}{2}}_{-\frac{1}{2}} (\eta_h) \right]^{-1} \nabla E_g \\
+ \frac{n}{T_e} \left[ 2 \frac{H^1_0 (\eta_e) + H^1_0 (\eta_h)}{H^{\frac{1}{2}}_{-\frac{1}{2}} (\eta_e) + H^{\frac{1}{2}}_{-\frac{1}{2}} (\eta_h)} - \frac{3}{2} \right] \nabla T_e \Bigg], \tag{4}
$$

where $H^\xi_\zeta (\eta_c) \equiv F_\xi (\eta_c) / F_\zeta (\eta_c)$ and the ambipolar diffusion coefficient is

$$D = \frac{k_B T_e}{q_e} \frac{\mu_e \mu_h H^0_{\frac{1}{2}} (\eta_e) H^0_{\frac{1}{2}} (\eta_h)}{\mu_e H^0_{\frac{1}{2}} (\eta_e) + \mu_h H^0_{\frac{1}{2}} (\eta_h)} \left[ H^{\frac{1}{2}}_{-\frac{1}{2}} (\eta_e) + H^{\frac{1}{2}}_{-\frac{1}{2}} (\eta_h) \right] \tag{5}$$

with $q_e$ the elementary charge. Ambipolar energy flow is the sum of diffusion and thermal energy currents inside the carrier subsystem and can be written as

$$\vec{W} = \left\{ E_g + 2 k_B T_e \left[ H^1_0 (\eta_e) + H^1_0 (\eta_h) \right] \right\} \vec{J} - (\kappa_e + \kappa_h) \nabla T_e. \tag{6}$$

The dynamics of semiconductors under the irradiation of ultrashort laser pulses can be modeled with the system of three continuum equations [17,28]: continuity equation for free carrier density and two coupled energy balance equations, one for the carriers and one for atoms:

$$\frac{\partial n}{\partial t} + \nabla \cdot \vec{J} = S_n - \gamma n^3 + \delta (T_e) \, n, \tag{7}$$

$$
\begin{aligned}
C_{e-h} \frac{\partial T_e}{\partial t} = &\, S_u - \nabla \cdot \vec{W} - \frac{C_{e-h}}{\tau_{e-p}} (T_e - T_a) \\
&- \frac{\partial n}{\partial t} \left\{ E_g + \frac{3}{2} k_B T_e \left[ H^{\frac{3}{2}}_{\frac{1}{2}} (\eta_e) + H^{\frac{3}{2}}_{\frac{1}{2}} (\eta_h) \right] \right\} - n \left( \frac{\partial E_g}{\partial n} \frac{\partial n}{\partial t} + \frac{\partial E_g}{\partial T_a} \frac{\partial T_a}{\partial t} \right) \\
&- \frac{3}{2} k_B T_e n \frac{\partial n}{\partial t} \left\{ \left[ 1 - H^{\frac{3}{2}}_{\frac{1}{2}} (\eta_e) H^{\frac{3}{2}}_{\frac{1}{2}} (\eta_e) \right] \frac{\partial \eta_e}{\partial n} + \left[ 1 - H^{\frac{3}{2}}_{\frac{1}{2}} (\eta_h) H^{-\frac{1}{2}}_{\frac{1}{2}} (\eta_h) \right] \frac{\partial \eta_h}{\partial n} \right\},
\end{aligned} \tag{8}
$$

$$C_a \frac{\partial T_a}{\partial t} = \nabla \cdot (k_a \nabla T_a) + \frac{C_{e-h}}{\tau_{ep}} (T_e - T_a). \tag{9}$$

The meanings of symbols in Equations (7) to (9) are the following, $S_n$ is the source of new carriers (excitation rate of new carriers by the laser), $S_u$ describes the energy source (rate of laser energy absorption), and $T_a$ is atomic temperature. The terms on the right hand side of Equation (7) are responsible for the carrier generation, Auger recombination, and impact ionization, consequently. Equation (8) describes the energy balance in the photoexcited electron–hole pairs and is a nonlinear

diffusion equation. The last Equation (9) describes the energy balance in the atomic subsystem. $C_{e-h}$ is specific heat capacity of the electron–hole pairs:

$$C_{e-h} = \frac{3}{2} n k_B \left[ H_{\frac{1}{2}}^{\frac{3}{2}} (\eta_e) + H_{\frac{1}{2}}^{\frac{3}{2}} (\eta_h) \right.$$

$$\left. + T_e \frac{\partial \eta_e}{\partial T_e} \left[ 1 - H_{\frac{1}{2}}^{\frac{3}{2}} (\eta_e) H_{\frac{1}{2}}^{-\frac{1}{2}} (\eta_e) \right] + T_e \frac{\partial \eta_h}{\partial T_e} \left[ 1 - H_{\frac{1}{2}}^{\frac{3}{2}} (\eta_h) H_{\frac{1}{2}}^{-\frac{1}{2}} (\eta_h) \right] \right]. \tag{10}$$

The total energy of electron–hole pairs consists of the energy gap and the kinetic energy of electrons and holes (taking into account the Fermi statistics),

$$u = n E_g (n, T_e) + \frac{3}{2} n k_B T_e \left[ H_{\frac{1}{2}}^{\frac{3}{2}} (\eta_e) + H_{\frac{1}{2}}^{\frac{3}{2}} (\eta_h) \right]. \tag{11}$$

The parameters used in the calculations as well as the meanings of other symbols are presented in Table 1.

To present an example of the model application, we use the following source terms. The rate of free carrier density growth, $S_n$, and the corresponding rate of their energy increase, $S_u$, are given by

$$S_n = \frac{\alpha I_{abs} (\vec{r}, t)}{\hbar \omega} + \frac{\beta I_{abs}^2 (\vec{r}, t)}{2 \hbar \omega}, \tag{12}$$

$$S_u = \alpha I_{abs} (\vec{r}, t) + \beta I_{abs}^2 (\vec{r}, t) + \Theta n I_{abs} (\vec{r}, t). \tag{13}$$

In the last two equations, the first and second terms on the right hand side represent the influence of one- and two-photon absorption, respectively, and the third term in the second equation represents the laser energy absorption by the excited free carriers.

One-dimensional (1D) laser heating problem is analyzed in this work. The laser is focused on the material surface. The corresponding form of laser intensity at the surface ($z = 0$) in this case is

$$I_{abs} (0, t) = (1 - R (T_a)) \sqrt{\frac{\varsigma}{\pi}} \frac{\Phi_{inc}}{t_p} \exp \left( -\varsigma \left[ (t - 3 t_p) / t_p \right]^2 \right), \tag{14}$$

where $\Phi_{inc}$ is the incident fluence, $\varsigma = 4 \ln 2$, and $R(T_a)$ is the reflectivity function (see Table 1). In the present calculations, the center of Gaussian pulse is shifted in time from the initial time, $t = 0$, to 3 pulse duration times, $3 t_p$, which in turn is defined as the pulse width at the half of maximum.

The dependence of the laser pulse intensity, $I_{abs}$, on depth can be found upon the solution of differential equation of the attenuation process:

$$\frac{d I_{abs}}{dz} = -\alpha I_{abs} (z, t) - \beta I_{abs}^2 (z, t) - \Theta n I_{abs} (z, t), \tag{15}$$

where $z$ is the depth into sample; the terms on the right side are responsible for one- and two-photon absorption, and for the free-carrier absorption processes.

Thus, from the system of Equations (7) to (9), we can fully determine the dynamics of $n$, $T_e$, and $T_a$ in 1D using the following initial and boundary conditions, suitable for a free standing film:

$$T_a (z, 0) = T_e (z, 0) = 300 \, \text{K},$$
$$n (z, 0) = n_{eq} = 1 \times 10^{16} \, \text{m}^{-3}, \text{ ref. [29]},$$
$$J (0, t) = J (L, t) = 0,$$
$$W (0, t) = W (L, t) = 0,$$
$$k_a \frac{\partial T_a}{\partial z} (0, t) = k_a \frac{\partial T_a}{\partial z} (L, t) = 0, \tag{16}$$

where $L$ is the thickness of the sample.

Owing to its similarity with an ordinary well-known TTM model [16], but with an additional equation for free carriers density $n$, we refer to the described approach as $n$TTM model, as it was suggested in [24].

**Table 1.** Model parameters.

| Parameter Name | Value | Citation |
|---|---|---|
| Initial carrier density | $n_0 = 1 \times 10^{16} \, \text{m}^{-3}$ | [29] |
| Initial lattice and carrier temperature | $T_0 = 300 \, \text{K}$ | |
| Lattice specific heat | $C_a = 1.978 \times 10^6 + 3.54 \times 10^2 T_a - 3.68 \times 10^6 / T_a^2$ , J/(m³K) ($T_a$ in K) | [30] |
| Lattice thermal conductivity | $k_a = 1.585 \times 10^5 \times T_a^{-1.23}$, W/(m·K) ($T_a$ in K) | [30] |
| Carrier thermal conductivity | $k_e = k_h = -3.47 \times 10^{18} + 4.45 \times 10^{16} T_e$, eV/(s m K) | [31] |
| Indirect band gap | $E_g = 1.170 - 4.73 \times 10^{-4} T_a^2 / (T_a + 636) - 1.5 \times 10^{-10} n^{1/3}$ if $1.170 - 4.73 \times 10^{-4} T_a^2 / (T_a + 636) - 1.5 \times 10^{-10} n^{1/3} \geq 0$ and 0 otherwise, eV ($T_a$ in K, $n$ in m⁻³) | [32] [33] |
| Interband absorption (taken from 694 nm laser) | $\alpha = 1.34 \times 10^5 \exp{(T_a/427)}$, m⁻¹ | [34] |
| Two-photon absorption | $\beta = 15 \, \text{cm/GW}$ | [25] |
| Reflectivity | $R = 0.329 + 5 \times 10^{-5}(T_a - 300)$ ($T_a$ in K) | [35] |
| Auger recombination coefficient | $\gamma = 3.8 \times 10^{-43}$, m⁶/s | [36] |
| Impact ionization coefficient | $\delta = 3.6 \times 10^{10} \exp{(-1.5 E_g / k_B T e)}$, s⁻¹ | [37] |
| Free-carrier absorption cross section | $\Theta = 2.91 \times 10^{-22} T_a / 300$, m² ($T_a$ in K) | [38] |
| Electron-phonon relaxation time | $\tau_{e-p} = 0.5 \times 10^{-12} \left[1 + n/(2 \times 10^{27})\right]$, s ($n$ in m⁻³) | [31] |
| Electron effective mass | $m_e^* = 0.36 m_e$ | [39] |
| Hole effective mass | $m_h^* = 0.81 m_e$ | [39] |
| Mobility of electrons (taken at 1000 K) | $\mu_e = 0.0085 \, \text{m}^2/\text{V·s}$ | [38] |
| Mobility of holes (taken at 1000 K) | $\mu_h = 0.0019 \, \text{m}^2/\text{V·s}$ | [38] |

## 3. Numerical Solution Scheme

To solve the system of Equations (7) to (9), we utilize the finite difference grid mesh presented in Figure 1. Sample is divided into cells as indicated, and the local thermodynamic parameters are calculated in each cell. The spatial derivatives of $n, T_e, T_a, J, W, k_a \frac{\partial T_a}{\partial z}$, and $E_g$ at the interior points are approximated with the central differences, and those at the boundaries are evaluated with the first-order approximation. Equations (7) and (9) are initially solved explicitly ($T \equiv T_e$):

$$\frac{n_i^{k+1} - n_i^k}{\Delta t} + \frac{J_i^k - J_{i-1}^k}{\Delta x} = (S_n)_i^k - \gamma(n_i^k)^3 + \delta_i^k n_i^k, \tag{17}$$

$$(C_a)_i^k \frac{(T_a)_i^{k+1} - (T_a)_i^k}{\Delta t} = \frac{1}{(\Delta z)^2} \left[ (k_a)_{i+\frac{1}{2}}^k (T_{i+1}^k - T_i^k) - (k_a)_{i-\frac{1}{2}}^k (T_i^k - T_{i-1}^k) \right]$$
$$+ \frac{(C_{e-h})_i^k}{(\tau_{e-p})_i^k} \left[ T_i^k - (T_a)_i^k \right],$$

(18)

where index $i$ is connected to the cell number (see Figure 1) and $k$ to the moment of time.

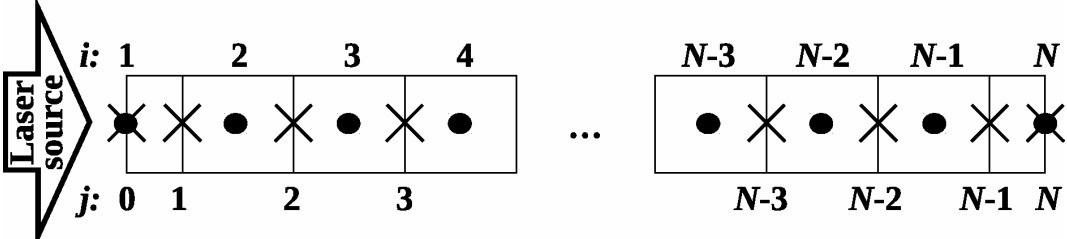

**Figure 1.** The finite-difference grid mesh for the solution of the *n*TTM model equations. Symbol "•" indicates the grid points for $n$, $T_e$, $T_a$, and $E_g$ ($i = 1, 2, ..., N$); symbol "×" indicates the grid points for $\vec{J}$, $\vec{W}$, $D$, and $k_a \frac{\partial T_a}{\partial z}$ ($j = 1, 2, ..., N+1$).

Therefore, before solving Equation (8) we already have the predictions for $n^{k+1}$ and $(T_a)^{k+1}$. The approach is based on the Crank–Nicolson semi-implicit scheme [21,22]. Equation (8) can be rewritten in the following finite-difference form:

$$\frac{T_i^{k+1} - T_i^k}{\Delta t} = (1 - \psi) f_i^k + \psi f_i^{k+1}.$$

(19)

The right-hand side contains parameter $\psi$, which can be 0 for explicit scheme, 1 for implicit, and $\frac{1}{2}$ for semi-implicit. The function $f_i^k$ can be found from:

$$(C_{e-h})_i^k f_i^k = (S_u)_i^k - \frac{W_i^k - W_{i-1}^k}{\Delta z} - \frac{(C_{e-h})_i^k}{(\tau_{e-p})_i^k} \left[ T_i^k - (T_a)_i^k \right]$$
$$- \left( \frac{\partial n}{\partial t} \right)_i^k \left\{ (E_g)_i^k + \frac{3}{2} k_B T_i^k \left[ H_{\frac{1}{2}}^{\frac{3}{2}}(\eta_e) + H_{\frac{1}{2}}^{\frac{3}{2}}(\eta_h) \right]_i^k \right\}$$
$$- n_i^k \left\{ \left( \frac{\partial E_g}{\partial n} \right)_i^k \left( \frac{\partial n}{\partial t} \right)_i^k + \left( \frac{\partial E_g}{\partial T_a} \right)_i^k \left( \frac{\partial T_a}{\partial t} \right)_i^k \right\}$$
$$- \frac{3}{2} k_B T_i^k n_i^k \left( \frac{\partial n}{\partial t} \right)_i^k \times \left\{ \left( \frac{\partial \eta_e}{\partial n} \right)_i^k \left[ 1 - \left( H_{\frac{1}{2}}^{\frac{3}{2}}(\eta_e) \right)_i^k \left( H_{\frac{1}{2}}^{-\frac{1}{2}}(\eta_e) \right)_i^k \right] \right.$$
$$\left. + \left( \frac{\partial \eta_h}{\partial n} \right)_i^k \left[ 1 - \left( H_{\frac{1}{2}}^{\frac{3}{2}}(\eta_h) \right)_i^k \left( H_{\frac{1}{2}}^{-\frac{1}{2}}(\eta_h) \right)_i^k \right] \right\},$$

(20)

where $W_i^k$ is defined according to Figure 1 (between the cells) and Equation (6):

$$W_i^k = \left\{ (E_g)_{i+\frac{1}{2}}^k + 2 k_B T_{i+\frac{1}{2}}^k [H_0^1(\eta_e) + H_0^1(\eta_h)]_{i+\frac{1}{2}}^k \right\} \times J_i^k$$
$$- (k_e + k_h)_{i+\frac{1}{2}}^k \frac{T_{i+1}^k - T_i^k}{\Delta z}$$

(21)

Analogously, according to Figure 1 and Equation (4), the carrier current is

$$
J_i^k = -D_i^k \left[ \frac{n_{i+1}^k - n_i^k}{\Delta z} + \frac{n_{i+\frac{1}{2}}^k}{k_B T_{i+\frac{1}{2}}^k} \left\{ \left[ H_{-\frac{1}{2}}^{\frac{1}{2}}(\eta_e) + H_{-\frac{1}{2}}^{\frac{1}{2}}(\eta_h) \right]_{i+\frac{1}{2}}^k \right\}^{-1} \frac{(E_g)_{i+1}^k - (E_g)_i^k}{\Delta z} \right.
$$

$$
\left. + \frac{n_{i+\frac{1}{2}}^k}{T_{i+\frac{1}{2}}^k} \left\{ 2 \frac{\left[ H_0^1(\eta_e) + H_0^1(\eta_h) \right]_{i+\frac{1}{2}}^k}{\left[ H_{\frac{1}{2}}^{-\frac{1}{2}}(\eta_e) + H_{\frac{1}{2}}^{-\frac{1}{2}}(\eta_h) \right]_{i+\frac{1}{2}}^k} - \frac{3}{2} \right\} \frac{T_{i+1}^k - T_i^k}{\Delta z} \right] \tag{22}
$$

with

$$
D_i^k = \frac{k_B T_{i+\frac{1}{2}}^k}{q_e} \frac{\mu_e \mu_h (H_{\frac{1}{2}}^0(\eta_e))_{i+\frac{1}{2}}^k (H_{\frac{1}{2}}^0(\eta_h))_{i+\frac{1}{2}}^k}{\mu_e (H_{\frac{1}{2}}^0(\eta_e))_{i+\frac{1}{2}}^k + \mu_h (H_{\frac{1}{2}}^0(\eta_h))_{i+\frac{1}{2}}^k} \left[ H_{-\frac{1}{2}}^{\frac{1}{2}}(\eta_e) + H_{-\frac{1}{2}}^{\frac{1}{2}}(\eta_h) \right]_{i+\frac{1}{2}}^k. \tag{23}
$$

Any function in between cells can be found by averaging

$$
A_{i+\frac{1}{2}} = \frac{1}{2}(A_i + A_{i+1}). \tag{24}
$$

The Fermi–Dirac integrals were calculated using GNU Scientific Library [40] and stored in the tables to speed up the calculations. $\frac{\partial \eta_c}{\partial n}$ can be found by derivative of Equation (2) by the carrier density:

$$
\left( \frac{\partial \eta_c}{\partial n} \right) = \frac{1}{2} \left( \frac{2\pi\hbar^2}{m_c^* k_B T_e} \right)^{\frac{3}{2}} \frac{1}{F_{\frac{1}{2}}(\eta_c)}; \tag{25}
$$

$\frac{\partial \eta_c}{\partial T_e}$ can be found by taking the derivative of equation Equation (2) by the electronic temperature:

$$
\left( \frac{\partial n_c}{\partial T_e} \right) = -\frac{3}{\sqrt{2}} n \times (T_e)^{-\frac{5}{2}} \left( \frac{\pi\hbar^2}{m_c^* k_B} \right)^{\frac{3}{2}} \frac{1}{F_{\frac{1}{2}}(\eta_c)}. \tag{26}
$$

The boundary conditions can be rewritten in the finite-difference form as follows,

$$
\frac{n_1^{k+1} - n_1^k}{\Delta t} + \frac{2 J_1^k}{\Delta z} = (S_n)_1^k - \gamma (n_1^k)^3 + \delta_1^k n_1^k; \tag{27}
$$

$$
(C_a)_1^k \frac{(T_a)_1^{k+1} - (T_a)_1^k}{\Delta t} = \frac{2}{(\Delta z)^2} (k_a)_{1+\frac{1}{2}}^k (T_2^k - T_1^k) + \frac{(C_{e-h})_1^k}{(\tau_{e-p})_1^k} \left[ T_1^k - (T_a)_1^k \right]; \tag{28}
$$

$$
\frac{T_1^{k+1} - T_1^k}{\Delta t} = (1 - \psi) f_1^k + \psi f_1^{k+1} \tag{29}
$$

with

$$
\begin{aligned}
(C_{e-h})_1^k f_1^k =& (S_u)_1^k - \frac{2W_1^k}{\Delta z} - \frac{(C_{e-h})_1^k}{(\tau_{e-p})_1^k} \left[ T_1^k - (T_a)_1^k \right] \\
& - \left( \frac{\partial n}{\partial t} \right)_1^k \left\{ (E_g)_1^k + \frac{3}{2} k_B T_1^k \left[ H_{\frac{1}{2}}^{\frac{3}{2}}(\eta_e) + H_{\frac{1}{2}}^{\frac{3}{2}}(\eta_h) \right]_1^k \right\} \\
& - n_1^k \left\{ \left( \frac{\partial E_g}{\partial n} \right)_1^k \left( \frac{\partial n}{\partial t} \right)_1^k + \left( \frac{\partial E_g}{\partial T_a} \right)_1^k \left( \frac{\partial T_a}{\partial t} \right)_1^k \right\} \\
& - \frac{3}{2} k_B T_1^k n_1^k \left( \frac{\partial n}{\partial t} \right)_1^k \times \left\{ \left( \frac{\partial \eta_e}{\partial n} \right)_1^k \left[ 1 - \left( H_{\frac{1}{2}}^{\frac{3}{2}}(\eta_e) \right)_1^k \left( H_{\frac{1}{2}}^{-\frac{1}{2}}(\eta_e) \right)_1^k \right] \right. \\
& \left. + \left( \frac{\partial \eta_h}{\partial n} \right)_1^k \left[ 1 - \left( H_{\frac{1}{2}}^{\frac{3}{2}}(\eta_h) \right)_1^k \left( H_{\frac{1}{2}}^{-\frac{1}{2}}(\eta_h) \right)_1^k \right] \right\};
\end{aligned}
\tag{30}
$$

and on the other edge

$$
\frac{n_N^{k+1} - n_N^k}{\Delta t} - \frac{2J_N^k}{\Delta z} = (S_n)_N^k - \gamma(n_N^k)^3 + \delta_N^k n_N^k;
\tag{31}
$$

$$
(C_a)_N^k \frac{(T_a)_N^{k+1} - (T_a)_N^k}{\Delta t} = \frac{-2}{(\Delta z)^2} (k_a)_{N-\frac{1}{2}}^k (T_N^k - T_{N-1}^k) + \frac{(C_{e-h})_N^k}{(\tau_{e-p})_N^k} \left[ T_N^k - (T_a)_N^k \right];
\tag{32}
$$

$$
\frac{T_N^{k+1} - T_N^k}{\Delta t} = (1 - \psi) f_N^k + \psi f_N^{k+1}
\tag{33}
$$

with

$$
\begin{aligned}
(C_{e-h})_N^k f_N^k =& (S_u)_N^k + \frac{2W_N^k}{\Delta z} - \frac{(C_{e-h})_N^k}{(\tau_{e-p})_N^k} \left[ T_N^k - (T_a)_N^k \right] \\
& - \left( \frac{\partial n}{\partial t} \right)_N^k \left\{ (E_g)_N^k + \frac{3}{2} k_B T_N^k \left[ H_{\frac{1}{2}}^{\frac{3}{2}}(\eta_e) + H_{\frac{1}{2}}^{\frac{3}{2}}(\eta_h) \right]_N^k \right\} \\
& - n_N^k \left\{ \left( \frac{\partial E_g}{\partial n} \right)_N^k \left( \frac{\partial n}{\partial t} \right)_N^k + \left( \frac{\partial E_g}{\partial T_a} \right)_N^k \left( \frac{\partial T_a}{\partial t} \right)_N^k \right\} \\
& - \frac{3}{2} k_B T_N^k n_N^k \left( \frac{\partial n}{\partial t} \right)_N^k \times \left\{ \left( \frac{\partial \eta_e}{\partial n} \right)_N^k \left[ 1 - \left( H_{\frac{1}{2}}^{\frac{3}{2}}(\eta_e) \right)_N^k \left( H_{\frac{1}{2}}^{-\frac{1}{2}}(\eta_e) \right)_N^k \right] \right. \\
& \left. + \left( \frac{\partial \eta_h}{\partial n} \right)_N^k \left[ 1 - \left( H_{\frac{1}{2}}^{\frac{3}{2}}(\eta_h) \right)_N^k \left( H_{\frac{1}{2}}^{-\frac{1}{2}}(\eta_h) \right)_N^k \right] \right\}.
\end{aligned}
\tag{34}
$$

All the other equations and connections between variables at the boundaries stay the same and can be straightforwardly obtained by substituting $i = 1$ and $i = N$ into Equations (17), (21), (25) and (26).

At the current time step, $k$, we do not have any information about the following parameters from the future time step $k + 1$,

$$
\begin{aligned}
& T_{i-1}^{k+1}, T_i^{k+1}, T_{i+1}^{k+1}, \left( \frac{\partial n}{\partial t} \right)_i^{k+1}, \left( \frac{\partial T_a}{\partial t} \right)_i^{k+1}, (S_u)_i^{k+1}, (C_{e-h})_i^{k+1}, J_i^{k+1}, \\
& \left( H_{\frac{1}{2}}^{\frac{3}{2}}(\eta_e) \right)_i^{k+1}, \left( H_{\frac{1}{2}}^{\frac{3}{2}}(\eta_h) \right)_i^{k+1}, \left( H_{\frac{1}{2}}^{-\frac{1}{2}}(\eta_e) \right)_i^{k+1}, \left( H_{\frac{1}{2}}^{-\frac{1}{2}}(\eta_h) \right)_i^{k+1}, \\
& \left( \frac{\partial \eta_e}{\partial n} \right)_i^{k+1}, \left( \frac{\partial \eta_h}{\partial n} \right)_i^{k+1}, \left( \frac{\partial \eta_e}{\partial T_e} \right)_i^{k+1}, \left( \frac{\partial \eta_h}{\partial T_e} \right)_i^{k+1}.
\end{aligned}
\tag{35}
$$

Initially we set them all except the first three to be equal to the corresponding old values (at time step $k$):

$$\left(A_i^{k+1}\right)^{(0)} = A_i^k. \tag{36}$$

Here, $(0)$ means the $0^{th}$ step of the corrector. With this assumption, Equation (19) becomes

$$a_i T_{i-1}^{k+1} + b_i T_i^{k+1} + c_i T_{i+1}^{k+1} = r_i, \tag{37}$$

which can be represented as a tridiagonal system of equations,

$$
\begin{bmatrix}
b_1 & c_1 & \cdots & & 0 \\
a_2 & b_2 & c_2 & & \\
 & a_3 & b_3 & & \cdots \\
\cdots & & & \cdots & c_{N-1} \\
0 & 0 & \cdots & a_N & b_N
\end{bmatrix}
\times
\begin{bmatrix}
T_1^{k+1} \\
T_2^{k+1} \\
\cdots \\
T_{N-1}^{k+1} \\
T_N^{k+1}
\end{bmatrix}
=
\begin{bmatrix}
r_1 \\
r_2 \\
\cdots \\
r_{N-1} \\
r_N
\end{bmatrix}, \tag{38}
$$

where

$$a_i = -\psi \frac{\Delta t}{\Delta z (C_{e-h})_i^{k+1}} \left( k_B [H_0^1(\eta_e) + H_0^1(\eta_h)]_{i-\frac{1}{2}}^{k+1} \times J_{i-1}^{k+1} + (k_e + k_h)_{i-\frac{1}{2}}^{k+1} / \Delta z \right), \tag{39}$$

$$
\begin{aligned}
b_i = 1 - \psi \frac{\Delta t}{(C_{e-h})_i^{k+1}} \times \Bigg( & -\frac{k_B}{\Delta z} \left[ H_0^1(\eta_e) + H_0^1(\eta_h) \right]_{i+\frac{1}{2}}^{k+1} \times J_i^{k+1} \\
& + \frac{k_B}{\Delta z} \left[ H_0^1(\eta_e) + H_0^1(\eta_h) \right]_{i-\frac{1}{2}}^{k+1} \times J_{i-1}^{k+1} \\
& - \left[ (k_e + k_h)_{i-\frac{1}{2}}^{k+1} + (k_e + k_h)_{i+\frac{1}{2}}^{k+1} \right] / (\Delta z)^2 - (C_{e-h})_i^{k+1} / (\tau_{e-p})_i^{k+1} \\
& - \frac{3}{2} \left( \frac{\partial n}{\partial t} \right)_i^{k+1} k_B \left( H_{\frac{1}{2}}^{\frac{3}{2}}(\eta_e) + H_{\frac{1}{2}}^{\frac{3}{2}}(\eta_h) \right)_i^{k+1} - \frac{3}{2} k_B n_i^{k+1} \left( \frac{\partial n}{\partial t} \right)_i^{k+1} \times \\
& \left\{ \left( \frac{\partial \eta_e}{\partial n} \right)_i^{k+1} \left[ 1 - \left( H_{\frac{1}{2}}^{\frac{3}{2}}(\eta_e) \right)_i^{k+1} \left( H_{\frac{1}{2}}^{-\frac{1}{2}}(\eta_e) \right)_i^{k+1} \right] \right. \\
& \left. + \left( \frac{\partial \eta_h}{\partial n} \right)_i^{k+1} \left[ 1 - \left( H_{\frac{1}{2}}^{\frac{3}{2}}(\eta_h) \right)_i^{k+1} \left( H_{\frac{1}{2}}^{-\frac{1}{2}}(\eta_h) \right)_i^{k+1} \right] \right\} \Bigg),
\end{aligned} \tag{40}
$$

$$c_i = \psi \frac{\Delta t}{\Delta z (C_{e-h})_i^{k+1}} \left( k_B [H_0^1(\eta_e) + H_0^1(\eta_h)]_{i+\frac{1}{2}}^{k+1} \times J_i^{k+1} - (k_e + k_h)_{i+\frac{1}{2}}^{k+1} / \Delta z \right), \tag{41}$$

$$
\begin{aligned}
r_i = T_i^k + (1 - \psi) \frac{\Delta t}{(C_{e-h})_i^k} f_i^k + \psi \frac{\Delta t}{(C_{e-h})_i^{k+1}} \times & \\
\Bigg( S_i^{k+1} - (E_g)_{i+\frac{1}{2}}^{k+1} \times \frac{J_i^{k+1}}{\Delta z} & + (E_g)_{i-\frac{1}{2}}^{k+1} \times \frac{J_{i-1}^{k+1}}{\Delta z} \\
+ \frac{(C_{e-h})_i^{k+1}}{(\tau_{e-p})_i^{k+1}} (T_a)_i^{k+1} & - \left( \frac{\partial n}{\partial t} \right)_i^{k+1} (E_g)_i^{k+1} \\
- n_i^{k+1} \left\{ \left( \frac{\partial E_g}{\partial n} \right)_i^{k+1} \left( \frac{\partial n}{\partial t} \right)_i^{k+1} & + \left( \frac{\partial E_g}{\partial T_a} \right)_i^{k+1} \left( \frac{\partial T_a}{\partial t} \right)_i^{k+1} \right\} \Bigg)
\end{aligned} \tag{42}
$$

for $i = 2, ..., N - 1$, and the boundary conditions are

$$
\begin{aligned}
b_1 = 1 - \psi \frac{\Delta t}{(C_{e-h})_1^{k+1}} \times \Bigg( & -\frac{k_B}{\Delta z} \left[ H_0^1(\eta_e) + H_0^1(\eta_h) \right]_1^{k+1} \times J_1^{k+1} \\
& - \frac{(k_e + k_h)_{\frac{3}{2}}^{k+1}}{(\Delta z)^2} - (C_{e-h})_1^{k+1} / (\tau_{e-p})_1^{k+1} \\
& - \frac{3}{2} k_B \left(\frac{\partial n}{\partial t}\right)_1^{k+1} \left( H_{\frac{1}{2}}^{\frac{3}{2}}(\eta_e) + H_{\frac{1}{2}}^{\frac{3}{2}}(\eta_h) \right)_1^{k+1} - \frac{3}{2} k_B n_1^{k+1} \left(\frac{\partial n}{\partial t}\right)_1^{k+1} \times \\
& \left\{ \left(\frac{\partial \eta_e}{\partial n}\right)_1^{k+1} \left[ 1 - \left( H_{\frac{1}{2}}^{\frac{3}{2}}(\eta_e) \right)_1^{k+1} \left( H_{\frac{1}{2}}^{-\frac{1}{2}}(\eta_e) \right)_1^{k+1} \right] \right. \\
& \left. + \left(\frac{\partial \eta_h}{\partial n}\right)_1^{k+1} \left[ 1 - \left( H_{\frac{1}{2}}^{\frac{3}{2}}(\eta_h) \right)_1^{k+1} \left( H_{\frac{1}{2}}^{-\frac{1}{2}}(\eta_h) \right)_1^{k+1} \right] \right\} \Bigg),
\end{aligned}
\tag{43}
$$

$$
c_1 = -\psi \frac{2\Delta t}{\Delta z (C_{e-h})_1^{k+1}} \left( -k_B [H_0^1(\eta_e) + H_0^1(\eta_h)]_{\frac{3}{2}}^{k+1} \times J_1^{k+1} + (k_e + k_h)_{\frac{3}{2}}^{k+1} / \Delta z \right),
\tag{44}
$$

$$
\begin{aligned}
r_1 = & T_1^k + (1 - \psi)\Delta t f_1^k + \psi \frac{\Delta t}{(C_{e-h})_1^{k+1}} \times \\
& \left( S_1^{k+1} - 2(E_g)_{\frac{3}{2}}^{k+1} \times \frac{J_1^{k+1}}{\Delta z} + \frac{(C_{e-h})_1^{k+1}}{(\tau_{e-p})_i 1^{k+1}} (T_a)_1^{k+1} - \left(\frac{\partial n}{\partial t}\right)_1^{k+1} (E_g)_1^{k+1} \right. \\
& \left. - n_1^{k+1} \left\{ \left(\frac{\partial E_g}{\partial n}\right)_1^{k+1} \left(\frac{\partial n}{\partial t}\right)_1^{k+1} + \left(\frac{\partial E_g}{\partial T_a}\right)_1^{k+1} \left(\frac{\partial T_a}{\partial t}\right)_1^{k+1} \right\} \right)
\end{aligned}
\tag{45}
$$

and

$$
\begin{aligned}
a_N = & -\psi \frac{\Delta t}{\Delta z (C_{e-h})_N^{k+1}} \times \\
& \left( 2k_B [H_0^1(\eta_e) + H_0^1(\eta_h)]_{N-\frac{1}{2}}^{k+1} \times J_{N-1}^{k+1} + (k_e + k_h)_{N-\frac{1}{2}}^{k+1} / \Delta z \right),
\end{aligned}
\tag{46}
$$

$$
\begin{aligned}
b_N = 1 - \psi \frac{\Delta t}{(C_{e-h})_N^{k+1}} \times \Bigg( & -2\frac{k_B}{\Delta z} \left[ H_0^1(\eta_e) + H_0^1(\eta_h) \right]_{N-\frac{1}{2}}^{k+1} \times J_{N-1}^{k+1} \\
& - \frac{2(k_e + k_h)_{N-\frac{1}{2}}^{k+1}}{(\Delta z)^2} - (C_{e-h})_N^{k+1} / (\tau_{e-p})_N^{k+1} \\
& - \frac{3}{2} k_B \left(\frac{\partial n}{\partial t}\right)_N^{k+1} \left( H_{\frac{1}{2}}^{\frac{3}{2}}(\eta_e) + H_{\frac{1}{2}}^{\frac{3}{2}}(\eta_h) \right)_N^{k+1} - \frac{3}{2} k_B n_N^{k+1} \left(\frac{\partial n}{\partial t}\right)_N^{k+1} \times \\
& \left\{ \left(\frac{\partial \eta_e}{\partial n}\right)_N^{k+1} \left[ 1 - \left( H_{\frac{1}{2}}^{\frac{3}{2}}(\eta_e) \right)_N^{k+1} \left( H_{\frac{1}{2}}^{-\frac{1}{2}}(\eta_e) \right)_N^{k+1} \right] \right. \\
& \left. + \left(\frac{\partial \eta_h}{\partial n}\right)_N^{k+1} \left[ 1 - \left( H_{\frac{1}{2}}^{\frac{3}{2}}(\eta_h) \right)_N^{k+1} \left( H_{\frac{1}{2}}^{-\frac{1}{2}}(\eta_h) \right)_N^{k+1} \right] \right\} \Bigg),
\end{aligned}
\tag{47}
$$

$$r_N = T_N^k + (1 - \psi)\Delta t f_N^k + \psi \frac{\Delta t}{(C_{e-h})_N^{k+1}} \times$$

$$\left( S_N^{k+1} + 2(E_g)_{N-\frac{1}{2}}^{k+1} \times \frac{J_{N-1}^{k+1}}{\Delta z} + \frac{(C_{e-h})_N^{k+1}}{(\tau_{e-p})_N^{k+1}} (T_a)_N^{k+1} - \left( \frac{\partial n}{\partial t} \right)_N^{k+1} (E_g)_N^{k+1} \right. \tag{48}$$

$$\left. - n_N^{k+1} \left\{ \left( \frac{\partial E_g}{\partial n} \right)_N^{k+1} \left( \frac{\partial n}{\partial t} \right)_N^{k+1} + \left( \frac{\partial E_g}{\partial T_a} \right)_N^{k+1} \left( \frac{\partial T_a}{\partial t} \right)_N^{k+1} \right\} \right).$$

Such a system can be resolved with respect to $\left\{ T_i^{k+1} \right\}_{i=1}^N$ by using the well-known tridiagonal matrix algorithm [41].

We denote the electronic temperature calculated with assumption (36) as $(T_i^{k+1})^{(1)}$, showing with "(1)" the first correction step. This result allows to calculate the corrected new values of functions in list (35):

$$[A_i^{k+1}]^{(1)} = A_i(T = [T_i^{k+1}]^{(1)}). \tag{49}$$

For an improved precision, the corrected new values of $n$ and $T_a$ can be calculated using the semi-implicit approach (instead of the explicit scheme, Equations (17) and (18), used for the predictor):

$$\left[ n_i^{k+1} \right]^{(1)} = n_i^k + (1 - \psi)\Delta t \left( \frac{\partial n}{\partial t} \right)_i^k + \psi \Delta t \left[ \left( \frac{\partial n}{\partial t} \right)_i^{k+1} \right]^{(1)}, \tag{50}$$

$$\left[ (T_a)_i^{k+1} \right]^{(1)} = (T_a)_i^k + (1 - \psi)\Delta t \left( \frac{\partial T_a}{\partial t} \right)_i^k + \psi \Delta t \left[ \left( \frac{\partial T_a}{\partial t} \right)_i^{k+1} \right]^{(1)}. \tag{51}$$

In turn, the corrected values allow to calculate $(T_i^{k+1})^{(2)}$ from Equation (38) and so on. Owing to its similarity with predictor–corrector methods, we call it a "predictor–corrector" algorithm. With this approach, Equation (19) can be rewritten in the following form,

$$(a_i)^{(l)}(T_{i-1}^{k+1})^{(l+1)} + (b_i)^{(l)}(T_i^{k+1})^{(l+1)} + (c_i)^{(l)}(T_{i+1}^{k+1})^{(l+1)} = (r_i)^{(l)}, \tag{52}$$
$$l = 0, 1, \dots,$$

where index $(l)$ shows the current step of correction and $(l) = (0)$ means the value is old, i.e., taken at time step $k$. This procedure continues until the difference between last two corrected values of electronic temperature is less than the demanded precision:

$$\sum_{i=1}^N \left[ (T_i^{k+1})^{(l+1)} - (T_i^{k+1})^{(l)} \right] < \varepsilon. \tag{53}$$

It takes up to 200 corrections to reach the chosen precision of $\varepsilon = 10^{-6}$ K during the laser pulse action, whereas when the laser is ended, 5 corrections are usually enough. The chosen numerical setup is discussed in the next section.

## 4. Calculation Example

As an example of application of our algorithm to the described system of equations Equations (7) to (9), we perform the simulations of 800 nm thick silicon target's response to ultrashort laser pulse irradiation. The parameters of the irradiation are 130 fs duration, 800 nm wavelength, and 0.26 J/cm$^2$ incident fluence. For these conditions, the experimental melting threshold fluence is 0.27 J/cm$^2$ [42], which is in agreement with the result of the $n$TTM model [43]. The value of fluence is chosen to be just below the melting threshold, providing the applicability of this simple model in the absence of phase transition processes. The sample was divided into 160 cells according to Figure 1. In Figure 2, we show the dynamics of electron–hole carrier density and electronic and atomic energy densities

at the silicon surface. The shown energy density is scaled to the melting energy density, which is found to be $3.86 \times 10^9$ J/m$^3$, according to the simulations. Though Equations (8) and (9) are written in terms of temperature of carriers and atoms, we plotted the corresponding energy densities instead, because their dynamics represents the energy flow between the subsystems and allows plotting the same scale for electrons and atoms, whereas electronic temperature is much higher than the atomic one (see also Figure 2 in [25]). In addition, this choice provides a possibility to show the energy conservation with the total average energy density of the sample (shown with black solid line).

The initial increase in the carrier density followed by the laser pulse is connected to the excitation of new carriers by one- and two-photon absorption processes. With time, the increase changes to the decay due to strong Auger recombination and diffusion processes. The strong peak in the electronic energy density is mostly connected to the free-carrier absorption. Finally, the thermal energy from electron–hole carriers is transferred to the atomic subsystem of the sample leading to gradual increase in the lattice energy density upon the electron–phonon equilibration.

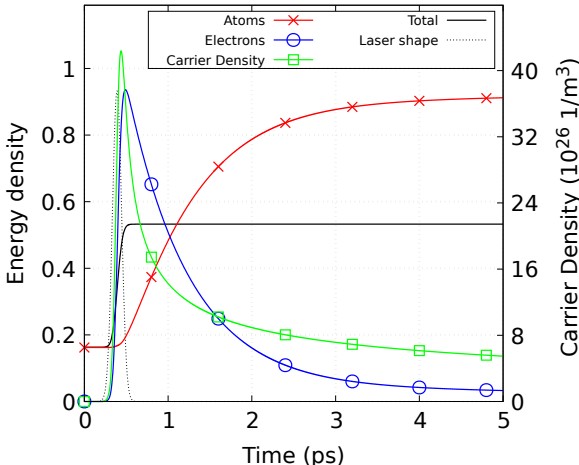

**Figure 2.** Electron/lattice energy densities (divided by the melting energy density) and carrier density dynamics, according to the *n*TTM model, at the surface of silicon target of 800 nm width, followed by the 130 fs laser pulse at the incident fluence of 0.26 J/cm$^2$. The total energy density, averaged through the whole sample, is shown with black solid line. The laser pulse shape, shown with black dotted line, is not in scale.

## 5. Discussion

In case of the explicit scheme, a good guess for the time step requirement can be obtained from the von Neumann stability criterion [20], $\Delta t \leq \frac{(\Delta x)^2}{2D_{th}}$, where $D_{th}$ is thermal diffusion coefficient, which is proportional to thermal conductivity $k_e$ and inversely proportional to the carrier heat capacity $C_{e-h}$. Under initial (prepulse) conditions, in the absence of free carriers, the latter tends to vanish (see Equation (10)), whereas the former is limited (see Table 1). After the laser irradiation starts, a quick increase of $T_e$ at initially low $n$ (see also Figure 2 in [25]) leads to an abrupt increase of $D_{th}$, which influences the von Neumann stability criterion and limits the maximum possible time step for explicit integration methods. Consequently, if one applies an explicit finite-difference scheme for the numerical solution, the stability of Equation (8) limits the maximum possible time step to $0.5 \times 10^{-19}$ s for the set of parameters published in [19]. Unfortunately, a mathematical error in [19] did not allow us to directly compare the results (specifically, in Equations (18) and (19) therein).

For the numerical setup presented here, according to our test calculations, the explicit scheme requires time steps as small as $1 \times 10^{-24}$ s, making calculations too expensive. In contrast, the proposed semi-implicit numerical integration scheme provides a stable solution for time step as high as $1 \times 10^{-16}$ s with the energy conservation about 0.16 % per simulation, Figure 3. At the same time, in case the calculation speed is critical, increasing the time step even higher is possible: $1 \times 10^{-15}$ s

provides the energy conservation within 1.6 %. Thus, the increase in the calculation speed of up to $10^9$ times has been achieved, compared with the explicit finite-difference integration scheme.

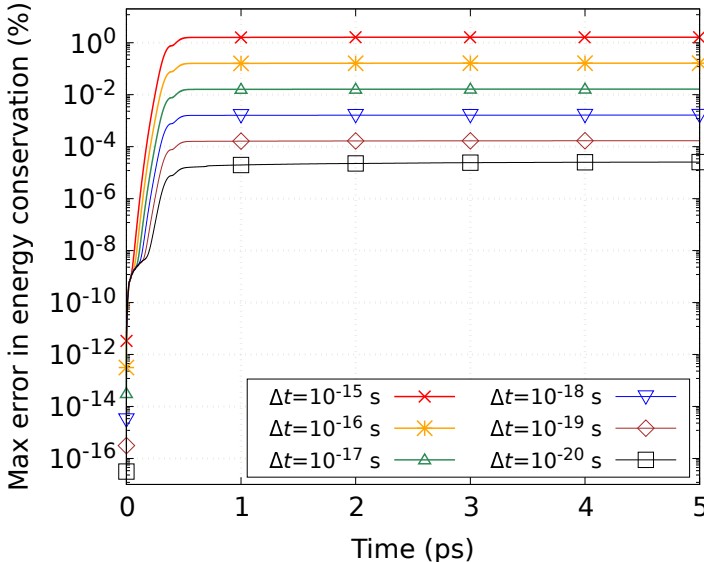

**Figure 3.** Maximal relative error in energy conservation as a function of simulation time for six different time steps. The quantity is calculated from the difference between total absorbed fluence and the total energy in the system.

Figure 4 shows the evolution of root mean square error (RMSE) in the electronic temperature for different time steps. The calculation with time step of $1 \times 10^{-20}$ s was used as a reference. Note that the chosen precision of the predictor–corrector ($10^{-6}$ K) is relatively low and likely influences RMSE at low time steps, preventing further improvement in RMSE for time steps below $\Delta t = 1 \times 10^{-17}$ s.

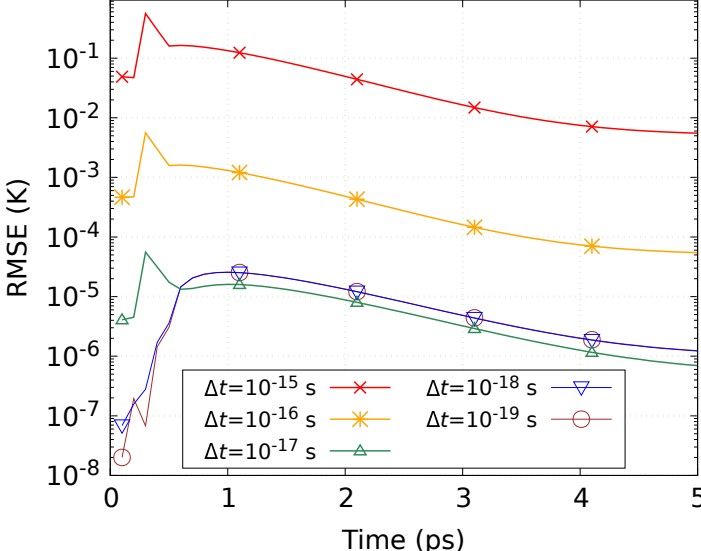

**Figure 4.** Root mean square error in electronic temperature depending on time for different time steps. The calculation with time step of $1 \times 10^{-20}$ s was used as a reference.

The time step is of course limited by the characteristic times of the involved physical processes, such as laser pulse duration, electron–phonon coupling time, and carrier recombination time. As long as it is much smaller than those mentioned above, the presented integration scheme is tested to be unconditionally stable.

This approach has been successfully applied earlier in order to investigate and improve the presented $n$TTM model [24]. In [25], we used the described scheme for the solution of the continuum part of the atomistic-continuum model MD-$n$TTM. The atomistic-continuum model describing the dynamics of gold targets under the ultrashort-pulse lasers also benefited from using the presented approach [23]. The high speed and precision of the scheme allowed to significantly decrease the computational costs of the corresponding simulations. Recently, Rathore et al. utilized the presented finite-difference scheme to simulate temporal evolution of photoinduced thermal strain in InSb [26].

In the mentioned applications, the corresponding system was solved in 1D, based on the assumption of wide laser spot in comparison with the lateral sizes of the computational setup [25]. Whenever it is not the case, one needs to solve the corresponding problem (the vector system of Equations (7) to (9)) in 2D or 3D case. According to the work in [44], the diffusion equation in 3D case can be solved in 3 subsequent steps, each of which involves implicit solution in only one direction ($X$, $Y$, or $Z$) and explicit scheme in the other two directions. In other words, one can use 1D implicit scheme three times: for $X$, $Y$, and $Z$ directions separately and consequently. This approach is called alternating direction implicit method (ADI) and is also widely applied for the corresponding 2D problems [45]. Therefore, with appropriate modifications, the presented scheme should be applicable for the considered nonlinear problem in 2D and 3D cases as well.

## 6. Conclusions

We proposed the semi-implicit integration scheme for the solution of nonlinear diffusion equations or systems of equations containing nonlinear diffusion equations. The scheme is based on the Crank–Nicolson finite-difference integration method, modified with a predictor–corrector algorithm, according to Equation (52). The modification resulted in a possibility to solve nonlinear diffusion equations with high stability and precision. The implemented Fortran source code is available in the Supplementary Materials under the terms of the GNU General Public License version 3 or later.

In the presented example of the scheme application, we reached the speed up of the calculations (by the increase of the integration time step) by up to $10^8$ times compared with the explicit scheme, keeping the error in energy conservation below 0.2 %. This error increases linearly with the time step. The algorithm is applicable in case the time step is much smaller than all the characteristic times of the involved physical processes. The existing applications that use the proposed scheme are mentioned and the possible extension for 2D and 3D cases is suggested.

**Supplementary Materials:** The following are available at http://www.mdpi.com/2076-3417/10/5/1853/s1.

**Author Contributions:** Conceptualization, D.I.; Data curation, V.L. and D.I.; Formal analysis, V.L. and D.I.; Funding acquisition, B.R., M.G., and D.I.; Investigation, V.L.; Methodology, V.L.; Project administration, B.R., M.G., and D.I.; Resources, B.R. and M.G.; Software, V.L. and D.I.; Supervision, B.R., M.G., and D.I.; Validation, V.L. and D.I.; Visualization, V.L.; Writing—original draft, V.L.; Writing—review & editing, V.L., B.R., M.G., and D.I. All authors have read and agreed to the published version of the manuscript

**Funding:** This work was supported by the Deutsche Forschungsgemeinschaft grants IV 122/1-1, IV 122/2, and RE 1141/15-1.

**Acknowledgments:** The authors acknowledge Markus Nießen for the assistance in the development of the described solution scheme. The work was partly conducted at the Institute for Laser Technology (ILT), RWTH, Aachen, Germany, department of "Nonlinear Dynamics of Laser Processing (NLD)", in the group of Wolfgang Schulz.

**Conflicts of Interest:** The authors declare no conflict of interest. The funders had no role in the design of the study; in the collection, analyses, or interpretation of data; in the writing of the manuscript; or in the decision to publish the results.

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
