# Peer review of "Solving a System of Differential Equations Containing a Diffusion Equation with Nonlinear Terms on the Example of Laser Heating in Silicon"

_applsci, doi:10.3390/app10051853_

Round 1

Reviewer 1 Report

The authors describe a semi-implicit numerical scheme to simulate laser interaction with semiconductor. They compare the standard explicit schem with their semi-implicit one.

Unfortunately, the only merit at the moment is the "total energy conservation" error.

The authors should provide more information on the scheme accuracy. They should plot the r.m.s. error of their new scheme for each relevant value as a function of the time step and compare it with the explicit scheme run at small time steps.

Author Response

Please see the attached response.

Reviewer 2 Report

The authors descibe a numerical scheme to solve the differential equations showing up in the two-temperature model applied to laser ablation. In the introduction, they go to length explaining that the method can be applied to many situations, but then they only discuss the TTM in a boring detailedness such that the reader misses the point completely.

To my opinion at least the title should be changed to reflect that his paper does not contain a general discussion of a method but its application to laser ablation.

Second I suggest that it is significantly simplified by concentrating really on the most important details of the TTM, and not presenteing all derivations.

Third, if it is kept as a general presentation, then at least one other application should be presented in some detail.

In principle I rather welcome the presentation of a numerical method, since this is often neglected and the reader cannot reproduce scientific results.

Thus I would recommend publication AFTER focusing the paper on the title and removing as much details about laser ablation as possible.

Author Response

Please see the attached response.

Round 2

Reviewer 2 Report

The authors have modified the manuscript removing the general statements and concentrating on the laser ablation simulation of silicon. The also adapted the title of the manuscript accoringly. As such they have fulfilled my concerns about the generality of their work.

On the other hand, I still have doubts if a reader would appreciate the discretized equations, although I often regret that even the flow diagram of a method is published nowhere.

If possible, I would encourage the authors to publish their implementation for example as a module for lammps.

At present I would recommend the publication of the manuscript although I find quite some overlap with previous publications by the authors.

Author Response

Please see pdf file attached.
